# The Potential of Korean Bioactive Substances and Functional Foods for Immune Enhancement

**DOI:** 10.3390/ijms25021334

**Published:** 2024-01-22

**Authors:** Mi Eun Kim, Jun Sik Lee

**Affiliations:** Department of Biological Science, Immunology Research Lab & BK21-Four Educational Research Group for Age-Associated Disorder Control Technology, Chosun University, Gwangju 61452, Republic of Korea; kimme0303@chosun.ac.kr

**Keywords:** Korean foods, functional foods, bioactive substances, immune enhancement

## Abstract

In this review, we explore the immunomodulatory properties of Korean foods, focusing on ginseng and fermented foods. One notable example is Korean red ginseng, known for its immune system-regulating effects attributed to the active ingredient, ginsenoside. Ginsenoside stimulates immune cells, enhancing immune function and suppressing inflammatory responses. With a long history, Korean red ginseng has demonstrated therapeutic effects against various diseases. Additionally, Korean fermented foods like kimchi, doenjang, chongkukjang, gochujang, vinegar, and jangajji provide diverse nutrients and bioactive substances, contributing to immune system enhancement. Moreover, traditional Korean natural herbs such as *Cirsium setidens* Nakai, Gomchwi, Beak-Jak-Yak, etc. possess immune-boosting properties and are used in various Korean foods. By incorporating these foods into one’s diet, one can strengthen their immune system, positively impacting their overall health and well-being.

## 1. Introduction

In the pursuit of optimizing health and well-being, an emerging field of research has focused on harnessing the immune-boosting properties of bioactive substances and functional foods, particularly those originating from Korea. This exploration is driven by a profound interest in traditional Korean knowledge and natural resources, which have been utilized for centuries to promote vitality and longevity.

In recent years, there has been a growing recognition of the central role that diet plays in shaping immune responses and overall health. Traditional Korean foods, known for their rich variety of flavors, ingredients, and cooking methods, have come under increasing scrutiny for their potential immunomodulatory properties [1,2,3,4,5]. In this review, we focus on the significant effects that Korean foods and bioactive ingredients may have on the immune system. A prominent element of Korean food is Korean red ginseng, which has a history of over 5000 years [6]. In particular, its active compounds, ginsenosides, have shown a remarkable ability to modulate the immune system. Ginsenosides stimulate immune cells and promote anti-inflammatory responses [7,8]. In this review, we explore the diversity of ginsenosides and their specific roles in enhancing immune function. Furthermore, the influence of Korean foods on the immune system extends far beyond ginseng. Moreover, beyond individual ingredients, Korean fermented foods like kimchi, doenjang, chongkukjang, and gochujang provide a unique array of nutrients and probiotics that profoundly impact the immune system [9,10]. Our investigation focuses on their roles in supporting gut health, regulating inflammation and immunity, and promoting the growth of beneficial gut bacteria.

In addition, Korean traditional herbs and vegetables, such as gomchwi, Beak-Jak-Yak, and ginger, serve as rich reservoirs of antioxidants, vitamins and minerals [11,12,13,14,15,16]. These antioxidants act as powerful agents against oxidative stress, thereby contributing significantly to immune function [17]. In summary, this comprehensive review attempts to elucidate the complex and multifaceted relationship between Korean foods and immune function. By meticulously scrutinizing the scientific evidence surrounding these dietary components, we aim to provide invaluable insights into the potential applications of traditional Korean foods and bioactive materials in enhancing human immunity. In doing so, we highlight the promising avenues for future research and the far-reaching implications for improving human health through conscious dietary choices.

## 2. Ginseng and Immunity

Korean red ginseng, derived from the root of *Panax ginseng*, is one of the most well-known functional foods in Korea. The ginsenosides, the plant’s active ingredients, have attracted considerable interest for their ability to modulate the immune system [8,18,19]. Studies have shown that they can stimulate immune cells, boost the body’s defenses and promote anti-inflammatory responses.

Ginseng, a traditional herb, has been used for over 5000 years for its potential health benefits. Initially, it was thought to have properties that could prolong human life. However, contemporary research has revealed its versatile therapeutic effects against a wide range of disorders, including physical ailments, viral diseases, and several types of cancer [8]. Several species of ginseng belong to the genus *Panax*, but *Panax ginseng*, also known as Korean ginseng, *P. quinquefolius*—American ginseng, *P. japonicus*—Japanese ginseng, and *P. notoginseng*—Sanqi ginseng stand out. Korean ginseng in particular has attracted a lot of interest and is widely used around the world [20,21]. Researchers have mainly focused on its roots, which come in several varieties, such as fresh, white, and red ginseng, depending on how much water they contain and how they have been processed. In particular, red ginseng undergoes a special processing stage involving incubation in 100 °C steam for more than two hours until its moisture content is reduced to less than 15% [8]. The primary bioactive constituents of ginseng, known as ginsenosides, are responsible for most of its medicinal effects. These ginsenosides have the prefix “Rx” before their names, where “x” indicates the polarity of the compound as measured by thin layer chromatography [8,22,23]. The least polar ginsenoside is known as Rh, while Ra is the most polar. Based on structural variations, ginsenosides can be further divided into four types: 20(*S*)-Protopanaxadiol (PPD), 20(*S*)-Protopanaxatriol (PPT), ocotillol and oleanolic acid types. Ginsenosides of the PPD type include Ra1, Ra2, Ra3, Rb1, Rb2, Rb3, Rc, Rd, F2, Rg3, Rg5, Rh2, Rh3, compound K, and PPD. Re, Rg1, Rg2, Rg4, Rh1, Rh4, Rf, and PPT form the PPT-type ginsenosides [8] (Figure 1). In addition, 24(*R*)-pseudoginsenoside-F11, 24(*R*)-pseudoginsenoside-RT5, majonoside R1, and R2 are among the chemicals that make up the ocotillol-type ginsenosides. Finally, Ro is a ginsenoside similar to oleanolic acid. The various therapeutic benefits associated with ginseng are a result of these ginsenosides, whose diverse structures and properties have stimulated continued study and interest in the fields of natural medicine and health sciences.

Ginseng has long been used as an immunomodulating herb. Ginseng can be used to boost human immunity and reduce inflammation caused by infections from external microbes [8,24,25,26,27]. Although ginseng has long been known for its immune-boosting properties, research is still underway to understand the complex mechanisms that underlie these effects. In order to understand the precise functions of different ginsenoside types, numerous studies have isolated and investigated these compounds [25,28,29,30]. In one of these studies, the immunomodulatory properties of heat-processed ginseng (HPG), including the Rg3, Rg5, and Rk1 complexes, were compared. Their research showed that HPG dramatically increases macrophage activation by interacting with the MAPK and NF-κB signaling pathways, primarily through the ERK and c-Jun pathway [28,31].

Consequently, cytokines such as TNF-a and IL-6 are produced as a result of this activation. The effect of ginsenoside Rg3 on macrophage phagocytosis was determined [8,32]. The activation of Rac1 and CDC42 and the upregulation of the phosphorylation of ERK and p38 MAPK, resulting in the stimulation of IL-6 and TNF-α, were observed to promote phagocytosis of IgG-opsonized Escherichia coli and IgG-opsonized beads. According to a comparison conducted by another researcher between fermented red ginseng (FRG) and red ginseng (RG), it was found that FRG significantly increased macrophage activity by activating the MAPK and NF-κB pathways [33]. By promoting the proliferation and production of TNF-α and IL-6, FRG also improved the immunostimulatory activity of central immune cells, including mouse spleen cells and bone marrow-derived macrophages. The innate immune system was found to be stimulated by the PPD-type ginsenoside 20S-dihydroprotopanaxadiol (2H-PPD) and compound K. By increasing phagocytic uptake, cell–cell aggregation, and surface levels of the costimulatory markers CD80 and CD86, these substances stimulate monocytes and macrophages [8,34,35]. While 2H-PPD does not promote cell adherence to the extracellular matrix and compound K reduces cell-tissue contacts, it also inhibits CD82, a hallmark of monocyte development. Compound K promotes TNF-α production via the NF-κB and AP-1 pathways [34]. Although the majority of in vitro studies have focused on PPD-type ginsenosides, certain PPT-type ginsenosides have also been investigated. The innate immune response in macrophages was found to be significantly enhanced by Rg1, a key component of PPT-type ginsenosides [36,37]. Rg1 also promoted the proliferation of CD3-, CD28-, and CD69-positive CD4+ T cells, which improved CD4+ T cell activity. In addition, Rg1 balanced Th1 and Th2 responses and treated pathogenic conditions characterized by Th1-dominant immune responses by suppressing IFN-γ and increasing IL-4 in isolated CD4+ T cells [8,38].

The immunostimulatory effects of various ginsenosides in ginseng have been demonstrated in a number of animal studies (Table 1). Several ginsenosides, including Rg1, Rg2, and Re, have shown the ability to potentiate the immune response when co-administered with specific antigens such as ovalbumin (OVA) and hepatitis B surface antigen (HBsAg) [8,39]. These ginsenosides have been shown to induce the production of antibodies, particularly IgG1 and IgG2a, the secretion of cytokines, including IL-4 and IFN-γ, and the activation of Th1 and Th2 immune cascades [24,40]. At the same time, other ginsenosides, including Rg3 and Rh2, have clearly demonstrated immunostimulatory properties. Rg3 has the potential to be important in the field of tumor immunity due to its ability to increase macrophage phagocytic activity and stimulate Th1 immune responses [41]. Rh2, on the other hand, has demonstrated the ability to enhance immune responses while minimizing chemotherapy-induced immunosuppression [42,43]. In addition, the PPD-type ginsenosides Rb1, Rb2, Rc, and Rd have demonstrated immunostimulatory effects by increasing antibody titers, cytokine production, and lymphocyte proliferation. A balanced Th1 and Th2 immunological homeostasis is partly orchestrated by these ginsenosides [8,44,45,46]. In conclusion, it has been shown that PPT and PPD ginsenosides have the ability to stimulate the immune system in a variety of animal models [8,41,47]. These ginsenosides show the ability to stimulate immunological responses when administered with specific antigens, making them potential candidates for use as adjuvants in the development of vaccines and immunotherapies (Figure 2). These findings have implications for improving human health and augmenting immunity in addition to illuminating the promising role of ginsenosides in enhancing and modifying the immune system.

## 3. Korean Fermented Foods

Korean fermented foods contain a wide range of properties that can help to improve the immune system. These foods provide a variety of nutrients and bioactive compounds, including probiotics, fermentation by-products, antioxidants, vitamins, minerals, and more, that help support the immune system. Here are some examples of how Korean fermented foods can boost the immune system.

### 3.1. Kimchi

Kimchi is a type of salted, fermented vegetable popular in Korea. It is low in calories (18 kcal/100 g), high in vitamins (such as beta-carotene, B-complex vitamins, and vitamin C), minerals (such as Ca, Fe, K, Na, and P), fiber (24% on a dry basis), and beneficial ingredients such as allyl compounds, capsaicin, chlorophyll, gingerol, and isothiocyanate. Indole compounds, thiocyanate, beta-sitosterol, and benzyl isothiocyanate are some of the active ingredients found in kimchi. These substances have been shown to have anti-cancer, anti-obesity, and anti-atherosclerotic activities [48]. In addition, the immune system is stimulated by the *Lactobacillus plantarum* (*L. Plantarum*) bacteria found in kimchi. Polysaccharides and cell wall fractions found in kimchi have immunogenic and mitogenic properties [49]. *L. plantarum* promotes the growth of immune cells, enhances the synthesis of antibodies, and increases the production of cytokines, including TNF-α and IL-6. Both enteric and systemic immune responses are improved by the oral administration of *L. plantarum* cell lysate. Kimchi’s *L. plantarum*, which stimulates macrophages, increases the production of NO, TNF-α and IL-6, and is essential for immunopotentiation activity, and also strengthens immunological function [49,50]. In addition, *L. plantarum* HY7712 has shown immunomodulatory effects, including a significant induction of NF-κB activation and TNF-α production. In mice with cyclophosphamide (CP)-induced immunosuppression, the oral administration of *L. plantarum* HY7712 resulted in the reversal of CP-induced effects, including changes in spleen and bone marrow cells, red and white blood cell levels, and body and spleen weights. These included changes in body and spleen weights, red blood and white blood cell levels, and changes in splenocytes and bone marrow cells. In particular, *L. plantarum* HY7712 reverses impaired macrophage phagocytosis and increases T-cell proliferation, IL-2, and IFN-γ production in cytotoxic T-cells, making it a potential immunotherapy [48,51].

### 3.2. Doenjang

Doenjang, a traditional fermented soybean paste from Korea, has been shown in numerous studies to have a wide range of potential health benefits. Antioxidant, fibrinolytic, antimutagenic, anticancer, and anti-obesity properties have all been noted [52,53,54]. In a preliminary study in mice, doenjang was found to reduce lipopolysaccharide (LPS) levels. Doenjang was then investigated for its effect on the composition of the gut microbiota in mice, and was found to increase *Bifidobacteria* while decreasing Enterobacteriaceae. Doenjang was also found to alter the dominant gut microbiota, with a decrease in Firmicutes and an increase in Bacteroidetes, and to reduce β-glucuronidase activity [52]. It also improved the levels of colonic tight junction proteins, particularly occludin. There was some suppression of NF-κB, but this was not statistically significant. In addition, doenjang showed potential to improve gut health by controlling gut microbiota and LPS levels while suppressing the production of damaging enzymes. It also significantly decreased tumor necrosis factor expression and increased interleukin-10 expression. In another study, doenjang powder was found to improve humoral, cellular, and pathogenic virus immunity in mice, including NK cell activation, potentially reducing the risk of influenza and atopic dermatitis. Another study investigated how doenjang prevented the development of ulcerative colitis (UC) in a mouse model treated with dextran sulphate sodium (DSS) [55]. Numerous doenjang samples were found to lengthen the colon, reduce UC signs and symptoms, and reduce immune-related blood indicators. They also reduced tissue damage while lowering the levels of pro-inflammatory cytokines such as TNF-α and IL-6. In addition, doenjang showed favorable effects on the gastrointestinal system according to an immunohistochemical study in mice. An increased expression of nitric oxide synthase, protein kinase C, and stem cell factor in the GI tract, increased CD4+/CD8+ lymphocytes in the jejunum and colon, and improved gastrointestinal motility, blood flow, and systemic immune activity were the results. These findings provide experimental evidence for the health benefits of the doenjang diet [52,53,54,55].

### 3.3. Chongkukjang

An important part of the traditional Korean foods, chongkukjang is a fermented soybean paste that is rich in nutrients, including high-quality proteins, essential amino acids, vitamins, minerals, and antioxidant compounds [9,56]. Due to the complex interactions between its components and the microbial changes that occur during the fermentation process, the consumption of chongkukjang has been associated with potential immunomodulatory effects. Amino acids are essential for the production of immune-related molecules such as antibodies and cytokines, which direct immune responses. In addition, the antioxidant properties of chongkukjang, due to elements such as flavonoids and polyphenols, provide a defense against oxidative stress and inflammation. By preventing cellular damage and reducing the stress that inflammation places on the immune system, these antioxidants support healthy immune function. Vitamin C, Vitamin D, and Zinc are three nutrients found in chongkukjang that are very important for immune health. With its antioxidant properties, vitamin C helps to scavenge free radicals and promote the activity of immune cells, and immune system compromise has been linked to vitamin D insufficiency, which plays a role in immune regulation [57]. Zinc, on the other hand, plays a role in the growth and function of immune cells, among other immune-related processes. A component of immune system improvement is chongkukjang’s potential to control inflammation. Chongkukjang’s potential to modify the inflammatory response may indirectly contribute to the improvement of immune function, as chronic inflammation is closely associated with immune dysfunction. Researchers have investigated the bioactive components in chongkukjang for their anti-inflammatory properties, which may reduce the inflammatory burden on the immune system and support immunological homeostasis. Furthermore, the effects of chongkukjang on immune function are linked to its effects on digestive health [58]. It is well known that a healthy gut microbiota, which is supported by the probiotics in chongkukjang, has a significant impact on the immune system. A healthy gut environment improves immune response and surveillance, while maintaining immunological tolerance. Thus, the consumption of chongkukjang can be seen as indirectly promoting immune function through its favorable effects on digestive health. In conclusion, chongkukjang, a fermented food rich in a variety of nutrients and bioactive compounds, has the potential to modulate the immune system through a number of mechanisms, including the promotion of probiotics, the provision of amino acids and antioxidants, and the support of essential vitamins and minerals. These immunomodulatory properties support the historical significance of chongkukjang as a healthy part of the Korean foods and call for further scientific research into its precise effects on immune function [9,57,58].

### 3.4. Gochujang

Gochujang, a traditional Korean fermented red pepper paste, may have immunological benefits due to its unique nutritional profile and the microbiological changes it undergoes during fermentation. Lactic acid bacteria and other beneficial microorganisms participate in the fermentation of gochujang, which has the potential to be probiotic [59]. These probiotics could help to modify the gut flora, which would then affect immunological responses. Immune homeostasis and resistance to infection depend on a healthy gut flora [60]. Gochujang contains a variety of bioactive compounds, including polyphenols, flavonoids, and capsaicin, the active ingredient in red pepper [61]. These substances act as antioxidants and immunomodulators. These substances have antioxidant and anti-inflammatory effects that can help to reduce chronic inflammation and oxidative stress, thereby improving immune function. Gochujang contains vitamins such as vitamin C and vitamin A, which are essential for the health of the immune system [61]. In particular, vitamin C supports immune cell activity and helps prevent infection. The immune system is also influenced by minerals such as potassium and iron. The capsaicin in red peppers may have a direct effect on immunological responses. Some research suggests that capsaicin may alter immune cell activity and promote anti-inflammatory effects, potentially improving overall immune function [62,63]. Immune health can benefit from a balanced gut flora, which is supported by the probiotics and fiber in gochujang. Immune cell growth and function are supported by a healthy gut environment, which also helps to prevent immune-related diseases. In addition, the complex mixture of gochujang influenced by fermentation and its key components may be a factor in its potential immunomodulatory effects. The regular consumption of gochujang as part of a healthy diet may help the immune system, but more research is needed to fully understand and measure its precise effects on immune function.

### 3.5. Vinegar

Vinegar, one of the oldest fermented foods in Korea, has historically been used as a condiment and in various home remedies. Known for its refreshing and sour taste that stimulates the appetite, vinegar is traditionally produced by natural fermentation using alcoholic beverages such as makgeolli, a rice-based alcoholic liquor [56,64]. Legal production also involves the dilution of synthetic acetic acid. Typically, production involves alcohol fermentation followed by acetic acid fermentation induced by the inoculation of acetic acid bacteria. Therefore, both naturally produced vinegar and acetic acid vinegar are used in Korean food. The existence and variety of uses of vinegar in Korea are confirmed by a multitude of historical documents. For instance, Gosacharlyo and Donguibogam claim that, during the Joseon Dynasty, vinegar fermented with barley was used to heal heartache, blood clots, carbuncles, and pain [56]. The early literature mentions a variety of vinegar varieties, and offers insights into the processes used in vinegar manufacturing. The main types of vinegar are grain vinegar and fruit vinegar, categorized according to the raw materials used. Although other types of vinegar exist, their consumption is limited. For example, vinegar made from medicinal herbs with additional ingredients has a limited local market. Raw materials such as brown rice, white rice, sorghum, barley, wheat, and millet require the saccharification process for vinegar production. Fruits, on the other hand, can be used directly for vinegar production. The type of vinegar is, therefore, determined by the primary ingredient in the production process. Probiotics such as Bacillus velezensis produced by this process have been reported to be used as antimicrobial agents against pathogenic infections such as *Klebsiella pneumoniae* [65].

### 3.6. Jangajji

Jangajji is Korean banchan (side dish) consisting of pickled vegetables in a soy sauce-based brine. Korea has a wide variety of pickles, some pickled in soy sauce, others in doenjang (Korean fermented soybean paste) or gochujang (Korean red chili paste). Among the different probiotics included in jangajji, *L. paraplantarum* SC61 showed higher antioxidant activity, such as DPPH radical scavenging, β-carotene discoloration inhibition, reducing power, superoxide anion scavenging, and ABTS radical scavenging activity. In addition, it was confirmed that *L. paraplantarum* SC61 significantly increased the mRNA expression levels of iNOS, IL-1β, IL-6, and TNF-α. Therefore, *L. paraplantarum* SC61 exhibits antioxidant and immunostimulatory activity, and has been shown to have potential use as a probiotic product [66]. Furthermore, in the case of green chili-pickled pepper (gochu-jangajji), significant DPPH free radical scavenging activity and superoxide reduction capacity were demonstrated in bovine mammary epithelial cells (MAC-T) cells [67]. As a traditional Korean food, pickles have a strong antioxidant capacity, and some foods have been found to possess immune-boosting activities.

## 4. Korean Traditional Natural Herbs

### 4.1. Gondre (Cirsium setidens Nakai)

*Cirsium setidens* Nakai (*C. setidens*) is a perennial herb of the Asteraceae family that grows only in Korea. The *C. setidens*, also known as ‘gondre,’ has uneven leaves and twigs on the edges of its leaves. The leaf tips are pointed, but the leaf base is quite broad, and the flowers are purple, blooming from July to October. In spring, the young shoots are picked and eaten as an herb, while the shoots are blanched, dried, and consumed as mung bean sprouts. In addition, *C. setidens* has been used in oriental medicine for various symptoms such as inflammation, swelling, and high blood pressure. Lee et al. have reported that *C. setides* extract has immunity enhancement efficacy. It not only inhibits the production of NO in macrophages, but also suppresses the gene expression of inflammatory factors such as IL-1β, IL-6, and TNF-α [68].

### 4.2. Gomchwi (Aster scaber)

Gomchwi is eaten all over Korea and is one of the most popular foods among Koreans. This gomchwi is used to make and eat chwinamul, which has various physiological and active functions. After being collected, the leaves and young stems are usually dried and cooked at a later time. It tastes pleasantly herbaceous when cooked, with a subtle, agreeable bitterness. According to Kim et al., investigations into cytokine production and splenocyte proliferation have indicated that gomchwi functions as an immunomodulator. In an in vivo experiment, gomchi water extract enhanced peritoneal macrophage TNF-α production and splenocyte proliferation [69]. Moreover, gomchwi extract showed a high inhibitory effect on DPPH and ABTS, and it was observed in murine alveolar macrophages that it not only suppressed the expression of proinflammatory diesters NO and COX-2, but also inhibited IL-1β and IL-6. Consequently, gomchwi demonstrated potent antioxidant and anti-inflammatory properties in light of these findings. These results also support the widespread culinary application of this highland plant due to its positive health properties [70].

### 4.3. Beak-Jak-Yak (Paeonia japonica var. Pilosa)

Beak-Jak-Yak is a medicinal plant which has been widely used as a component of blood-building decoctions in Korea. According to a report by Kim et al., it was observed that when Beak-Jak-Yak water extract was administered orally, the production ability of IL-1β, IL-6, and TNF-α increased in activated peritoneal macrophages. Based on these findings, Beak-Jak-Yak may improve the immunological function in mice by controlling splenocyte proliferation and their ability to produce cytokines [15]. In addition, Beak-Jak-Yak extract increased NO and cytokine production in LPS-stimulated macrophages [14]. Therefore, Beak-Jak-Yak is used in various herbal medicines in Korea and is consumed as a tea.

### 4.4. Ginger (Zingiber officinale Roscoe)

Ginger (*Zingiber officinale* Roscoe), which belongings to the Zingiber genus of the *Zingiberaceae* family, has been commonly consumed as a spice and herbal medicine in Korea for a long time. Ginger contains a variety of bioactive substances, including phenolic and terpene chemicals. The primary phenolic chemicals responsible for the different bioactivities of ginger are gingerols, shogaols, and paradols. It has been reported that gingerol increases the expression of NO and TNF-α, as well as myeloperoxidase in animal experiments, while shogaol inhibits the production of NO and PGE_2_ in macrophages [71]. In addition, ginger extract has been reported to be effective in boosting immunity by regulating the production of IFN-γ and IL-10 in immune cells [16,72]. In Korea, ginger, with its various immune-boosting effects, is consumed and enjoyed in various forms of food.

## 5. Conclusions

Building on previous studies that have demonstrated the immune-enhancing properties of various Korean foods, this study focuses specifically on the immune-enhancing properties of traditional Korean foods. In addition, the mechanisms underlying the immunomodulatory effects of Korean red ginseng, medicinal herbs, and fermented foods were explored, as well as how these elements interact with the immune system. The most important part of the immune-enhancing effects of Korean foods is the modulation of immune-enhancing factors and the positive effects on the gut microbiome. Traditional fermented Korean foods have been shown to enhance intestinal immunity through various beneficial bacteria in the gut, which in turn leads to immune-enhancing effects. This review not only synthesizes the existing body of knowledge, but also highlights critical gaps in our understanding, signaling the need for further investigation into the potential synergies and therapeutic applications of these dietary components. By contextualizing these findings within the broader landscape of nutritional immunology, we shed light on how traditional Korean foods can be integrated into personalized dietary strategies to improve immune health. In conclusion, our academic exploration of the immunomodulatory potential of Korean foods serves as a springboard for further scientific investigation and offers a promising avenue for the development of dietary recommendations and interventions that can benefit public health on a global scale.

## Figures and Tables

**Figure 1 ijms-25-01334-f001:**
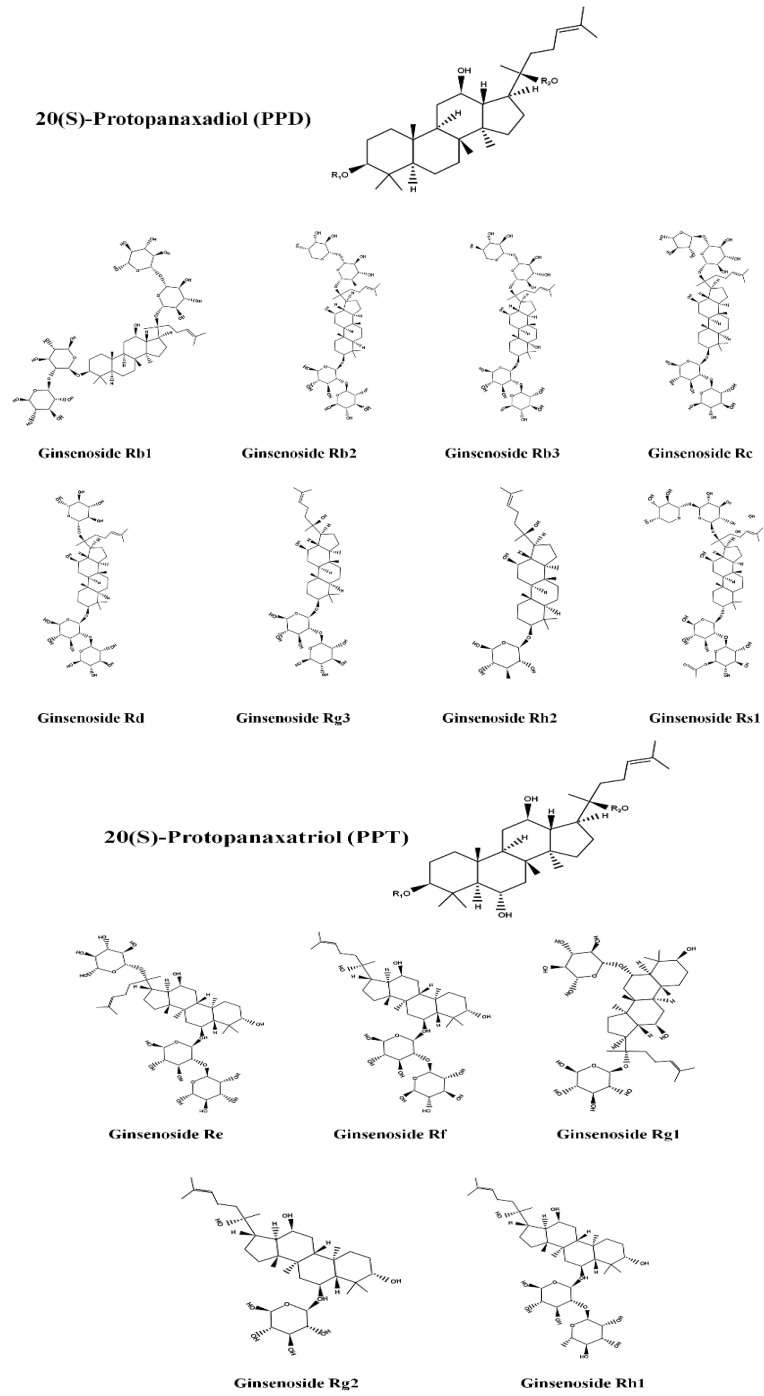
Major ginsenoside 20(S)-protopanaxadiol (PPD) and 20(S)-protopanaxatriol (PPT) structural features. The chemical structure was drawn using ChemDraw Pro 8.0 software.

**Figure 2 ijms-25-01334-f002:**
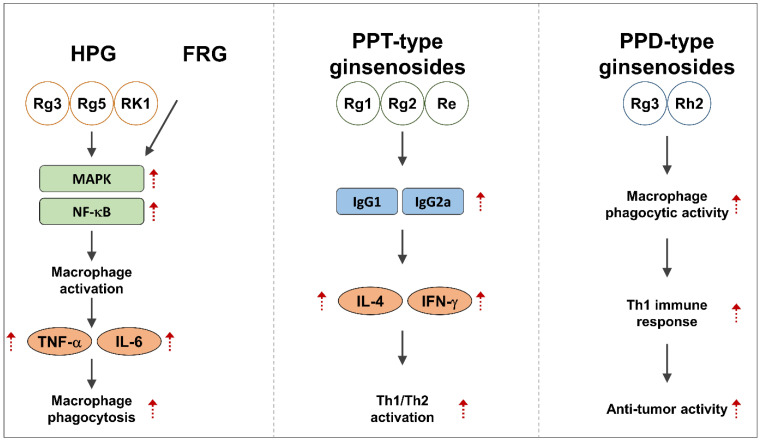
The immunomodulatory efficacy and regulatory mechanisms of ginsenosides. FRG, fermented red ginseng; HPG, heat-treated ginseng; IFN, interferon; Ig, immunoglobulin; IL, interleukin; NF-κB, nuclear factor kappa B; MAPK, mitogen-activated protein kinase; PPD, Protopanaxadiol; PPT, Protopanaxatriol; TNF, tumor necrosis factor.

**Table 1 ijms-25-01334-t001:** Summary of immunological efficacy of ginseng and ginsenosides.

Ginseng and Ginsenosides	Immunological Functions	Refs.
Korean Red Ginseng	- Activation and enhancement of immune cells - Promotion of anti-inflammatory responses	[8,20]
Rg3 (PPT-type ginsenoside)	- Increased activation of macrophages - Stimulation of inflammatory cytokines (TNF-α, IL-6) production through MAPK/AP-1 and NF-κB signaling pathways - Enhancement of macrophage phagocytic activity	[8,29,33]
Rg1 (PPT-type ginsenoside)	- Increased activation of macrophages and CD4+ T cells - Upregulation of TNF-α production via PI3K/Akt and mTOR pathways, while downregulating the NF-κB pathway leading to decreased IL-6 release - Balance of Th1 and Th2 immune responses	[39,40]
Rg2 (PPT-type ginsenoside)	- Promotion of immune responses - Anti-inflammatory effects - Potential for anti-cancer activity	[8,40]
Re (PPT-type ginsenoside)	- Promotion of immune responses - Anti-inflammatory effects - Potential for anti-cancer activity	[25,40]
Rb1, Rb2, Rc, Rd (PPD-type ginsenosides)	- Increased antibody levels - Enhanced cytokine production - Stimulation of lymphocyte proliferation - Maintenance of Th1 and Th2 immune balance	[45,46,47,48]

## Data Availability

The data presented in this study are available on request from the corresponding author. The data are not publicly available due to privacy.

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
