# Peer review of "The Potential of Korean Bioactive Substances and Functional Foods for Immune Enhancement"

_ijms, 2024, doi:10.3390/ijms25021334_

Round 1

Reviewer 1 Report (New Reviewer)

Comments and Suggestions for Authors

Kim et al. tried to provide an overview of the immune system enhancement by Korean functional foods. The manuscript is of interest to the reader however some improvements are required as follows:

1- The chemical structures of secondary metabolites such as ginsenoside derivatives should be drawn.

2- The plant or bacterial Genus and species should be in italics throughout the manuscript. There are also mistakes such as (Lab. Plantarum). Lactobacillus plantarum is abbreviated as L. plantarum

3- The authors are encouraged for some systems biology analyses to find the target genes that are potentially affected by kimchi compounds such as capsaicin, chlorophyll, gingerol, and isothiocyanate.

4- The conclusion of the manuscript should focus on the most important immune system components that are affected by Korean foods.

Comments on the Quality of English Language

Moderate English editing is required.

Author Response

Response for Reviewer I

Kim et al. tried to provide an overview of the immune system enhancement by Korean functional foods. The manuscript is of interest to the reader however some improvements are required as follows:

Comment 1: The chemical structures of secondary metabolites such as ginsenoside derivatives should be drawn.

Response: Thank you for your helpful comment. As requested by the reviewer, we have drawn the chemical structures of secondary metabolites such as ginsenoside derivatives and included them in Figure 1.

Comment 2: The plant or bacterial Genus and species should be in italics throughout the manuscript. There are also mistakes such as (Lab. Plantarum). Lactobacillus plantarum is abbreviated as L. plantarum.

Response: Thank you for your helpful comment. We have made the changes requested by the reviewer.

Comment 3: The authors are encouraged for some systems biology analyses to find the target genes that are potentially affected by kimchi compounds such as capsaicin, chlorophyll, gingerol, and isothiocyanate.

Response: Thank you for your helpful comment. Targeted genetic studies on the anti-cancer and anti-obesity effects of compounds such as capsaicin, chlorophyll, gingerol, and isothiocyanate in kimchi are currently ongoing, but there is not enough data for a comprehensive review. This review focuses on the beneficial effects of these compounds on the gut microbiome and the observed immune-enhancing effects.

Comment 4: The conclusion of the manuscript should focus on the most important immune system components that are affected by Korean foods.

Response: Thank you for your helpful comment. We included a mention of important immune system components affected by Korean foods in the Conclusion section

Reviewer 2 Report (New Reviewer)

Comments and Suggestions for Authors

Readers were more intrigued by the manuscript containing Korean natural botanicals and red ginseng. They provided additional details in this manuscript regarding bioactive substances found in Korean. In general, I believe that this work is suitable for publication with a few minor comments.

Line no 55: Can you explain the mechanism of how Korean Red ginseng acts as an oxidative stress agent?

It would be appreciated if you could include your whole article and add extra figures to your work.

How many different varieties of ginseng are present in Korea?

Please provide some further information regarding the Kimchi section.

Author Response

Response for Reviewer II

Readers were more intrigued by the manuscript containing Korean natural botanicals and red ginseng. They provided additional details in this manuscript regarding bioactive substances found in Korean. In general, I believe that this work is suitable for publication with a few minor comments.

Comment 1: Line no 55: Can you explain the mechanism of how Korean Red ginseng acts as an oxidative stress agent?

Response: Thank you for your helpful comment. Korean red ginseng (Panax ginseng) is often suggested to have antioxidant properties, which means it can help combat oxidative stress. Oxidative stress occurs when there is an imbalance between the production of reactive oxygen species (ROS) and the body's ability to counteract or detoxify their harmful effects through neutralization by antioxidants. In addition, Korean red ginseng can stimulate the activity of various antioxidant enzymes within cells, such as superoxide dismutase (SOD), catalase and glutathione peroxidase. These enzymes play a crucial role in neutralizing reactive oxygen species and maintaining cellular redox balance. Korean red ginseng may also affect oxidative stress through various signaling pathways, including the nuclear factor erythroid 2-related factor 2 (Nrf2) pathway, the phosphoinositide 3-kinase (PI3K)/Akt pathway, the mitogen-activated protein kinase (MAPK) pathway, the AMP-activated protein kinase (AMPK) pathway, and HO-1 induction. For example, MAPK pathways, including ERK (extracellular signal-regulated kinase), JNK (c-Jun N-terminal kinase), and p38 MAPK, are involved in cellular responses to oxidative stress and inflammation. Korean red ginseng has been reported to influence these pathways, potentially mitigating oxidative damage. In addition, Korean red ginseng may induce the expression of heme oxygenase-1, an enzyme with antioxidant and anti-inflammatory properties. HO-1 plays a role in cellular defense against oxidative stress.

Comment 2: It would be appreciated if you could include your whole article and add extra figures to your work.

Response: Thank you for your helpful comment. We have made the changes requested by the reviewer.

Comment 3: How many different varieties of ginseng are present in Korea?

Response: Thank you for your helpful comment. There are several types of ginsengs grown in Korea. Panax ginseng, Panax notoginseng and Panax quinquefolius are cultivated. Among them, Panax ginseng is the most famous and commonly cultivated type of ginseng used in Korean food. This review is based on the efficacy of Panax ginseng.

Comment 4: Please provide some further information regarding the Kimchi section.

Response: Thank you for your helpful comment. We have added additional information about the kimchi section, as requested by a reviewer.

Round 2

Reviewer 1 Report (New Reviewer)

Comments and Suggestions for Authors

The response to comment 1 is not convincing. Instead of using the PubChem compounds the authors should draw the chemical structures by themselves using software such as ChemDraw.

Comments on the Quality of English Language

Moderate English modification is required.

Author Response

Response for Reviewer I

Comment 1: The response to comment 1 is not convincing. Instead of using the PubChem compounds the authors should draw the chemical structures by themselves using software such as ChemDraw.

Response: Thank you for your helpful comment. As requested by the reviewer, the structure of ginsenosides was drawn in ChemDraw.

This manuscript is a resubmission of an earlier submission. The following is a list of the peer review reports and author responses from that submission.

Round 1

Reviewer 1 Report

Comments and Suggestions for Authors

The review entitled “The Potential of Korean Bioactive Substances and Functional Foods for Immune Enhancement” aims at discussing the immunomodulatory effects of the bioactives therein. However, in my opinion the review significantly lacks a thorough literature review on the topic. Some specific comments I have are:

1.       The abstract is vague and needs to be reframed to highlight the content of the paper

2.       Several papers remain uncited in this manuscript. For instance, there are several reports on Korean red ginseng that point at the immunomodulatory effects of the bioactive compounds/fractions therein Eg: https://doi.org/10.3390/molecules25163569. There are several other papers which despite being a few years older than the current time frame considered for this review, but are still crucial for the discussed topic and must be cited. If the authors decide to stick with the most recent work, I suggest that the review paper be resubmitted as a mini review.

3.       Despite the fact that the foods cited in the review paper such as Kimchi, Doenjang, Chongkukjang and Gochujang are specific to Korea, the mushrooms that have been discussed are mostly found in Europe and parts of China, amongst other regions of the world. Discussing them as “Korean functional foods” does not do justice to the topic being discussed.

4.       Agaricus blazei has been misspelled in the section title.

5.       The section Antioxidants from Korena foods is vague and includes fruits/foods that are found all over the world. The section is poorly discussed and does not add value to the paper. The table in this section also lacks appropriate references.

6.       All the tables in the manuscript lack references

7.       The importance of including the structures of ginsenosides in Figure 1 is unclear.

8.       Several fermented foods which could have potential probiotics that impact the immune system are missing.

Comments on the Quality of English Language

The English language is good and might only need minor edits

Author Response

Response for Reviewer I

The review entitled “The Potential of Korean Bioactive Substances and Functional Foods for Immune Enhancement” aims at discussing the immunomodulatory effects of the bioactives therein. However, in my opinion the review significantly lacks a thorough literature review on the topic. Some specific comments I have are:

Comment 1: The abstract is vague and needs to be reframed to highlight the content of the paper

Response: Thank you for your helpful comment. We have modified the parts of the abstract presented by the reviewer and we have created a new abstract.

Comment 2: Several papers remain uncited in this manuscript. For instance, there are several reports on Korean red ginseng that point at the immunomodulatory effects of the bioactive compounds/fractions therein Eg: https://doi.org/10.3390/molecules25163569. There are several other papers which despite being a few years older than the current time frame considered for this review, but are still crucial for the discussed topic and must be cited. If the authors decide to stick with the most recent work, I suggest that the review paper be resubmitted as a mini review.

Response: Thank you for your helpful comment. Following the reviewer's comments, we have added references from discussions of important earlier papers.

Comment 3: Despite the fact that the foods cited in the review paper such as Kimchi, Doenjang, Chongkukjang and Gochujang are specific to Korea, the mushrooms that have been discussed are mostly found in Europe and parts of China, amongst other regions of the world. Discussing them as “Korean functional foods” does not do justice to the topic being discussed.

Response: Thank you for your helpful comment. As the reviewer suggests, these mushrooms were first cultivated in Europe or other countries, but all of these mushrooms are cultivated and used for food in Korea. Furthermore, mushrooms are used and enjoyed as functional food in Korea. If the reviewer doesn't think it's relevant to the discussion, feel free to leave this section out, but we'd be happy to discuss it as a Korean functional food since these mushrooms are also grown in Korea.

Comment 4: Agaricus blazei has been misspelled in the section title

Response: Thank you for your helpful comment. We fixed it.

Comment 5: The section Antioxidants from Korena foods is vague and includes fruits/foods that are found all over the world. The section is poorly discussed and does not add value to the paper. The table in this section also lacks appropriate references.

Response: Thank you for your helpful comment. As per the reviewer's request, we have incorporated information on the antioxidant capacity of Korean food into our paper, providing relevant references throughout.

Comment 6: All the tables in the manuscript lack references

Response: Thank you for your helpful comment. We added all references to all tables.

Comment 7: The importance of including the structures of ginsenosides in Figure 1 is unclear.

Response: Thank you for your helpful comment. Based on the comments made by the reviewer, we excluded the structural drawings of the ginsenosides.

Comment 8: Several fermented foods which could have potential probiotics that impact the immune system are missing.

Response: Thank you for your helpful comment. In response to the reviewer's comments, we have made contributions to the paper by adding sentences about the efficacy of fermented foods in sections 3.5 and 3.6.

Reviewer 2 Report

Comments and Suggestions for Authors

This review delves into the multifaceted domain of traditional Korean cuisine and its impact on immune system functionality. The current research encompasses a diverse array of dietary components, including Korean red ginseng, medicinal mushrooms, and fermented foods, all of which have exhibited intriguing immunomodulatory characteristics. The authors elucidate the intricate interactions between these components and the immune system, using a substantial body of research and emphasizing their potential for synergistic effects and therapeutic utility. According to a range of research findings, it has been proposed that traditional Korean diets have the potential to positively impact immunological health within the wider framework of nutritional immunology.

This review covered all the areas of Korean Bioactive substances in the field of study and it added comprehensive information to this field of study.

I would suggest having a couple of cartoons to summarize the active ingredients and their immunological functions in Korean fruits and vegetables. It should cover the active ingredients, targets, and outcomes of particular inflammatory pathways or mechanisms.

In tables 1 and 2, add the citations for each immunological function/enhancement covered.

Author Response

Response for Reviewer II

This review delves into the multifaceted domain of traditional Korean cuisine and its impact on immune system functionality. The current research encompasses a diverse array of dietary components, including Korean red ginseng, medicinal mushrooms, and fermented foods, all of which have exhibited intriguing immunomodulatory characteristics. The authors elucidate the intricate interactions between these components and the immune system, using a substantial body of research and emphasizing their potential for synergistic effects and therapeutic utility. According to a range of research findings, it has been proposed that traditional Korean diets have the potential to positively impact immunological health within the wider framework of nutritional immunology.

This review covered all the areas of Korean Bioactive substances in the field of study and it added comprehensive information to this field of study.

Comment 1: I would suggest having a couple of cartoons to summarize the active ingredients and their immunological functions in Korean fruits and vegetables. It should cover the active ingredients, targets, and outcomes of particular inflammatory pathways or mechanisms.

Response: Thank you for your helpful comment. In response to a reviewer's comment, we have included figures on the immunomodulatory mechanisms of Korean Foods (Figures 1 and 2).

Comment 2: In tables 1 and 2, add the citations for each immunological function/enhancement covered.

Response: Thank you for your helpful comment. Following the reviewer's comments, we have added references.

Round 2

Reviewer 1 Report

Comments and Suggestions for Authors

In my opinion, the review still lacks a thorough literature review on the topic and is not suitable to be published as a full-length review. As the authors admit, the mushrooms discussed are not native to Korea, and discussing them in a paper entitled "Korean Functional Foods" is incorrect. Apart from that, the manuscript does not have enough content to be published as a full-length review paper. Antioxidants being discussed as functional bioactives do not add any value to a review paper. There are several bioactives that are unique to foods exclusively found in a particular geographic region, which should ideally make up the content of a well-written quality review on functional foods and bioactives.

Comments on the Quality of English Language

The English language could be improved.

Reviewer 2 Report

Comments and Suggestions for Authors

Authors sufficiently improved the manuscript and adequately answered my comments. No further comments.